# Sodium Promoted FeZn@SiO₂-C Catalysts for Sustainable Production of Low Olefins by CO₂ Hydrogenation

**Zhijiang Ni \*, Mingxing Cai, Shiyu Zhong, Xiaoyu Chen, Hanyu Shen and Lin Su**

School of Mechanical Engineering & Urban Rail Transit, Changzhou University, Changzhou 213001, China; cmingxing@126.com (M.C.)

\* Correspondence: nizhijiang@cczu.edu.cn or nizhijiang@126.com

**Abstract:** A prepared FeZnNa@SiO₂-C catalyst with graphitized carbon (C)-modified mesoporous SiO₂ supports metal nanoparticles with the sol–gel method. The effect of adding metal Na and Zn promoters as a dispersion on the CO₂ hydrogenation to low olefins was systematically studied. The results showed that Zn–Na, as a combination, could promote the absorption of CO₂ and improved the conversion rate of CO₂. Na as an alkaline substance can improve the absorption of more acidic CO₂, which could increase the conversion rate of CO₂ to 59.03%. Meanwhile, the addition of secondary metal Zn to Fe-based catalysts to form a surface alloy could alter the adsorption of CO₂ and the activation of C-O bonds, inhibit the subsequent hydrogenation of olefins to paraffins, and facilitate the reduction of Fe₂O₃ and the formation of active Fe₅C₂ species. The formation of active Fe₅C₂ species was found in TEM and XRD, and the selectivity of the target product was 41.07%. The deep hydrogenation of olefins was inhibited, and the space–time yield (STY) of low olefins was raised again by inhibiting their deep hydrogenations, up to 0.0436. However, the corresponding STY did not increase infinitely with the increase of Na doping, and higher catalytic performance for CO₂ hydrogenation could be exhibited when the Na doping reached 6.4%. Compared with Fe@SiO₂-C catalyst, Na- and Zn-promoted Fe-based catalysts, prepared by the modified sol-gel method, can be used directly for highly efficient CO₂ hydrogenation to low olefins and thus has a more promising application prospect in the future.

**Keywords:** FeZnNa@SiO₂-C; promoters effects; low olefins; CO₂ hydrogenation

## 1. Introduction

Increasing carbon dioxide emissions (CO₂) have a significant impact on climate change. CO₂ capture and utilization has received much attention as a viable pathway towards a low carbon economy [1–4]. CO₂ hydrogenation is a promising route to convert CO₂ into value-added C₂₊ hydrocarbons, such as olefins, aromatics, and gasoline, and has attracted industry and academic research worldwide [4–10]. Among them, the process of CO₂ hydrogenation to olefins should be divided into two steps. Firstly, CO intermediates are formed by the Reverse Water Gas Shift (RWGS) reaction and then hydrogenated to hydrocarbons by the Fischer–Tropsch Synthesis (FTS) reaction [11].

Compared to other catalysts [12–14], iron-based catalysts remain a promising system for the hydrogenation of CO₂ to low olefins due to their high reverse water–gas shift activity, flexible operating conditions and low cost [15]. However, the competitive carburization of a single iron phase in the catalyst and the evolvement into active phase Fe₅C₂ are not efficient. Therefore, the addition of other promoters could improve the structural evolution of the iron phase to increase the selectivity of olefin and long-chain hydrocarbons, and the study of the effect of promoters on the dynamic transformation of the iron phase is important for the development of efficient catalysts.

To date, alkaline species (AMs, M = Na [16–19] and K [20–22]) and transition metals (TMs, M = Ni, Co, Cr and Mn) [23–25] have been commonly introduced into Fe-based

catalysts to compete carburizing and the evolution into active phase $Fe_5C_2$ for improving the selectivity of olefins. In particular, the addition of Na promotes the ability of $Fe_5C_2$ formation as the interaction between Na and Fe species inhibited the hydrogenation of olefins by increasing the electron density of the active sites, leading to a higher $CO_2$ conversion rate and improved olefin selectivity [26,27]. Yao [18] and Zhang [11] reconfirmed that the low olefins/paraffins (O/P) and $CO_2$ conversion rate were significantly increased by the addition of the adjuvant Na doping. More recently, co-modification with TMs and AMs is gaining high popularity as an emerging strategy to accelerate $CO_2$ dissociative adsorption and improve iron carbide phase formation for favor olefin synthesis [28]. However, the effects of secondary metal and Na promoters on catalyst structures and performances of $CO_2$ hydrogenation remain ambiguous. For instance, the Zn promoter is usually considered as the structure promoter, but it has also been proposed as an electronic promoter or the active phase in the form of ZnO for RWGS. Furthermore, the studies on the synergistic effects of Zn and Na promoters were seldom in the previous literature.

Combined with our previous work [29,30], GC-modified mesoporous silica-encapsulated metal nanoparticles can effectively improve reducibility and electron conductivity properties similar to those of expensive noble metals (Pt [31] and Ru [32]). In this study, $FeZnNa@SiO_2$-C catalysts were prepared by the sol–gel method via regulating the amount of Na loading. The role of these Zn and Na promoters was rigorously analyzed by decoupling the rates of the RWGS and FTS steps and considering the proximity to the reaction equilibrium. In addition, a combination of quasi in situ structural characterization measurements, including X-ray photoelectron spectroscopy and Transmission Electron Microscope (TEM), etc., were used to establish a clear structure–property relationship for $CO_2$ hydrogenation on Fe-based catalysts. During this process, synergetic promoting effects by Zn and Na are elucidated, which were responsible for the concurrently enhanced $CO_2$ conversion rates and olefin selectivity. These results would contribute to the rational regulation of catalytic performance by multiple additives, especially for complex reaction systems that are generally encountered in heterogeneous catalysis.

## 2. Results and Discussion

### 2.1. Characterization of the Fe-Based Catalysts

2.1.1. Textural Property, Structure and Morphology

As shown in Figure 1a,b, the calcined $SiO_2$-C, $Zn@SiO_2$-C, $Na@SiO_2$-C, $Fe@SiO_2$-C and $FeNa@SiO_2$-C catalysts exhibit the characteristic diffraction patterns of the $Fe_2O_3$ phase (JCPDS No. 39-1346) and $SiO_2$ phase (JCPDS No. 39-1425), while the $Zn@SiO_2$-C and $FeZnNa_x@SiO_2$-C catalysts provide diffraction peaks mainly assigned to $ZnFe_2O_4$ spinel phases (JCPDS No. 22-1012), in addition to the weak signals for the ZnO (JCPDS No. 36-1451) phase. The specific results are shown in Figure 1a,b and Table 1. The particle size of $Fe_2O_3$ in $FeZn@SiO_2$-C catalysts, evaluated by the Scherrer equation ($D = K \times \gamma/(B \times \cos\theta)$ formula [33], means that K, B, θ, γ are the Scherrer constant, the half peak width or integral width of the measured sample diffraction peak, and the Prague angle, which was ~15 nm, respectively, much smaller than those of the $Fe@SiO_2$-C (~23 nm), $FeNa@SiO_2$-C (~33 nm) and $FeZnNax@SiO_2$-C (~20 nm). These results indicate that the Zn promoter significantly decreases the size of Fe species over Fe catalysts [11]. However, the addition of Na changed the particle size of the phase in the $FeZnNa_x@SiO_2$-C catalysts. The particle size of $Fe_2O_3$ generally showed a trend of first increasing and then decreasing with the increase in Na doping. Xiong [34] reported that the introduction of alkali metals promoted the agglomeration of iron particles, thereby increasing the size of iron oxide particles. The presence of alkali metals appears to promote the agglomeration of the iron precursor, thus enlarging the crystallite size of the iron oxide. In conjunction with Figure 1b, no $Na_2O$ (JCPDS No. 03-1074) phase was found in $FeNa_x@SiO_2$-C and $FeZnNa_x@SiO_2$-C samples, which is mainly due to the low Na doping [11]. Moreover, it is also possible that there are no XRD patterns associated with the $Na_2O$/Na phase, which suggests a high dispersion of $Na_2O$/Na in these catalysts [18]. Figure 1b shows that the diffraction peaks

of ZnO (36.25° and 34.42°) are more obvious when the Na doping in FeNa$_x$@SiO$_2$-C and FeZnNa$_x$@SiO$_2$-C samples reached 6.4%. It can be found that the specific surface area (SSA) of FeZnNa$_x$@SiO$_2$-C catalysts increased from 9.261 m$^2$/g to 20.572 m$^2$/g as the particle size of Fe$_2$O$_3$ decreases. Combined with the larger specific surface area of the remaining catalysts and the final actual CO$_2$ conversion rate, it can be found that although the remaining catalysts have a larger specific surface area, the actual CO$_2$ conversion rate is lower than that of any of FeZnNa$_x$@SiO$_2$-C catalysts, which shows that Zn and Na as promoters can overcome the disadvantage of its own smaller specific surface area and obtain a more practical advantage of higher CO$_2$ conversion rate. If we only consider alkaline Na as a promoter of the absorption of acidic CO$_2$, it can be found that there is still a clear difference between their CO$_2$ conversion rate by comparing FeZnNa$_x$@SiO$_2$-C catalysts. Therefore, the increase in CO$_2$ conversion rate was still attributed to the bifunctional effect of Zn and Na.

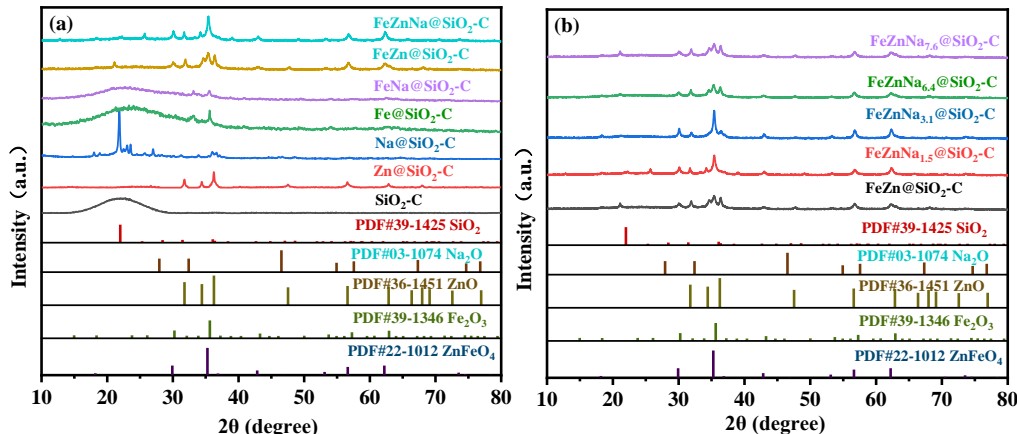

**Figure 1.** XRD pattern of (**a**) SiO$_2$-C, Zn@SiO$_2$-C, Na@SiO$_2$-C, Fe@SiO$_2$-C, FeNa@SiO$_2$-C and FeZn@SiO$_2$-C catalysts, (**b**) FeZnNa$_x$@SiO$_2$-C catalysts with different Na doping.

**Table 1.** Physicochemical characteristics of the Fe@SiO$_2$-C, FeNa@SiO$_2$-C and FeZnNa$_x$@SiO$_2$-C catalysts.

| Catalyst | Fe Loading (wt%) [a] | Na Loading (wt%) [a] | Total CO Chemisorbed (μmol/g) [b] | % Fe Dispersion [b] | Fe$_2$O$_3$ Size (nm) [c] | SSA (m$^2$/g) [d] | PV (cm$^3$/g) [d] | APS (nm) [d] |
|---|---|---|---|---|---|---|---|---|
| Fe@SiO$_2$-C | 25.4 | 0 | 472 | 5.2 | 23 | 75.5 | 0.087 | 3.19 |
| FeNa@SiO$_2$-C | 24.2 | 1.5 | 570 | 6.6 | 33 | 78.4 | 0.111 | 4.18 |
| FeZn@SiO$_2$-C | 25.3 | 0 | 551 | 6.1 | 15 | 48.5 | 0.144 | 4.19 |
| FeZnNa$_{1.5}$@SiO$_2$-C | 23.7 | 1.5 | 576 | 6.8 | 21 | 9.2 | 0.035 | 4.46 |
| FeZnNa$_{3.1}$@SiO$_2$-C | 23.5 | 3.1 | 596 | 7.1 | 23 | 16.4 | 0.035 | 2.95 |
| FeZnNa$_{6.4}$@SiO$_2$-C | 22.1 | 6.4 | 576 | 7.3 | 16 | 20.5 | 0.047 | 2.95 |
| FeZnNa$_{7.6}$@SiO$_2$-C | 23.4 | 7.6 | 568 | 6.8 | 18 | 14.6 | 0.035 | 2.95 |

[a] Measured by XRF; [b] Based on total CO chemisorbed (corrected), $n_{CO}/n_{Fe}$ = 0.5, % dispersion = $2 \times n_{\text{total CO chemisorbed}} / n_{\text{total number of Fe atoms}}$; [c] Measured with the Scherrer equation by XRD; [d] Measured by N$_2$ physical adsorption.

Figure 2 shows the TEM images and high-resolution TEM images of FeZnNa$_x$@SiO$_2$-C; it can be determined that the FeZnNa$_x$@SiO$_2$-C series catalysts have a mesoporous structure with a large number of particles distributed. The nanoparticles are not easy to produce carbon deposits with critical particle size. The small particle size is attributed to the fact that P123 is rich in hydroxyl groups (-OH), which are able to bind tightly with TEOS molecules to form micelles. The C overlay produced by polymer carbonation inhibits the sintering of metal nanoparticles and metal, and support interaction is enhanced during high-temperature treatment. The well-dispersed Fe$_2$O$_3$ nanoparticles, with an average diameter of 15–21 nm on Fe$_2$O$_3$, can be clearly observed, as shown in Figure 2. The lattice spacing of the Fe$_2$O$_3$ particle on SiO$_2$ is 0.26 nm, which can be attributed to the (400) planes, suggesting that the catalyst exists in the form of Fe$_2$O$_3$ and retains good crystallinity, consistent with the XRD results. Figure 2f,h shows that there are two different oxides,

Fe$_2$O$_3$ and ZnO. These metal particles have a very small particle size and are dispersed on mesoporous SiO$_2$, including Fe$_2$O$_3$ and ZnO or Na$_2$O/Na. In the reduction step, these metal oxides are reduced to Fe$_2$O$_3$/ZnO at a temperature of 673 K. In the catalytic process, more active Fe$_5$C$_2$ particles are generated to improve the selectivity of the target product.

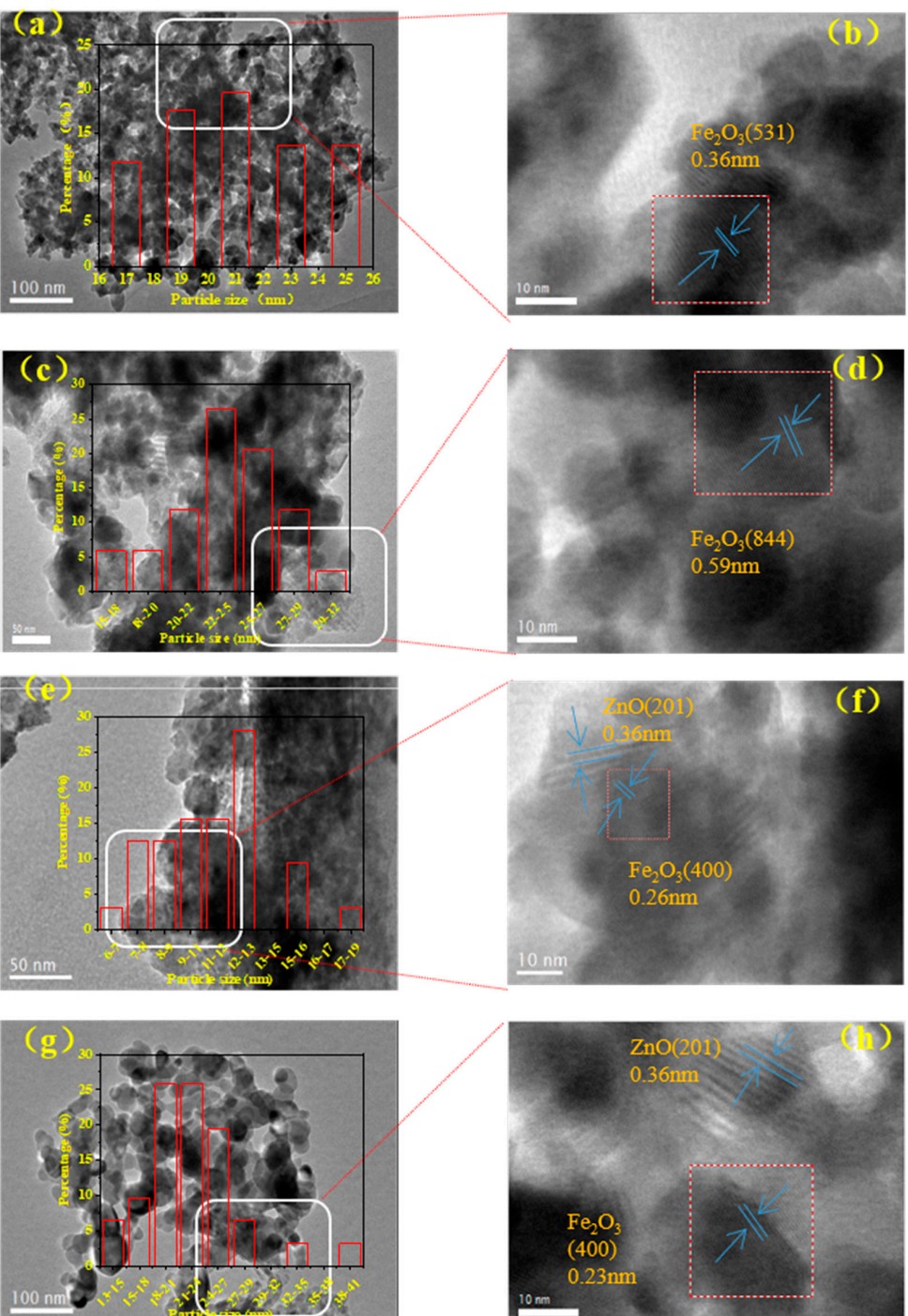

**Figure 2.** TEM images and High-resolution TEM images of (**a**,**b**) FeZnNa$_{1.5}$@SiO$_2$-C, (**c**,**d**) FeZnNa$_{3.1}$@SiO$_2$-C, (**e**,**f**) FeZnNa$_{6.4}$@SiO$_2$-C, (**g**,**h**) FeZnNa$_{7.6}$@SiO$_2$-C.

Apparently, in Figure 3, the FeZnNa$_x$@SiO$_2$-C nanoparticles give a type-IV isotherm with a long hysteresis loop at relative pressure (P/P$_0$) of 0.4–1.0, which is characteristic of mesoporous materials according to the classification of International Union of Pure and Applied Chemistry (IUPAC). As shown in Table 1, the specific surface area generally

shows a trend of first increasing and then decreasing with the increase in Na doping; FeZnNa$_{6.4}$@SiO$_2$-C had the highest specific surface area compared with other catalysts at 20.572 m$^2$/g. Liang [35] found that when the Na doping was 0.1%, 0.5% and 5%, respectively, the specific surface area also increased first and then decreased, which were 31, 45, 28 m$^2$/g. Although the specific surface area of the other three catalysts was higher than that of FeZnNa$_{6.4}$@SiO$_2$-C catalysts, the actual CO$_2$ conversion rate was lower than that of FeZnNa$_{6.4}$@SiO$_2$-C catalysts, which indicated that ZnNa can significantly improve the CO$_2$ conversion rate. Herein, the change in conversion rate is also consistent with Liang's study [35] that alkaline additives can promote the adsorption and conversion of CO$_2$.

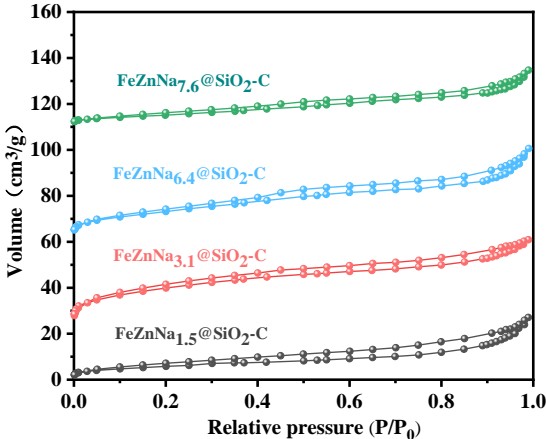

**Figure 3.** N$_2$ adsorption–desorption curves of the FeZnNa$_x$@SiO$_2$-C catalysts with the different Na doping.

The results are consistent with the related reports that an Na promoter will increase the CO$_2$ adsorption and adsorption strength. Therefore, the adsorption of CO$_2$ on the surface of the FeZnNa$_x$@SiO$_2$-C catalyst is stronger than other catalysts, and it is easier to activate CO$_2$, which is conducive to the subsequent conversion. The pore volume of FeZnNa$_{6.4}$@SiO$_2$-C also reached the maximum value of 0.047 cm$^3$/g. The average pore size reached a maximum at 1.5%, and the remaining three groups of catalysts did not differ much, and still achieved the maximum value of 2.956 nm at 6.4%, and the pore volume decreased from 0.047 cm$^3$/g at 6.4%Na to 0.035 cm$^3$/g at 7.6% Na. However, the change in pore volume was very small, which showed that the excessive increase in Na doping did not have much effect on the volume of the pore. However, the pores of the catalyst do got clogged by excess Na. When the Na doping were 1.5%, 3.1 and 6.4%, which was not excessive, the pores were not clogged. Therefore, the decrease in surface area is suggested to be due to crystallite size growth rather than pore clogging [34].

### 2.1.2. XPS Results

The valence states of components in catalysts were investigated by XPS, the FeZn@SiO$_2$-C catalyst delivered a Fe 2p3/2 peak at the binding energy of 711.23 eV and a satellite peak around 719.07 eV, which was slightly higher than the corresponding peak of the standard Fe$_2$O$_3$ sample, which confirmed the existence of the Fe species in a form of Fe$^{3+}$ species. The Fe 2p3/2 binding energy of the FeZnNa1.5@SiO$_2$-C catalyst in Figure 4a was reduced by about ~0.18 eV compared to that of the FeZn@SiO$_2$-C catalyst, which indicates that the introduction of Na promoter reduced the Fe 2p3/2 binding energy of the FeZnNa$_{1.5}$@SiO$_2$-C catalyst. The decrease in binding energy suggests the charge transfer from Na ions to the vacant d orbits of Fe species [11]. Among them, the existence of C-O bonds indicates that there may be close interactions between C-Fe$_2$O$_3$-SiO$_2$ in the form of C-O-Fe/Fe-C-O or C-O-Si (Figure 4b). The O/Si ratio is higher than that of 2.0 for SiO$_2$, indicating that it may be due to the enhanced interaction between C and Fe nanoparticles and SiO$_2$. The satellite peak is approximately 718.81 eV, it is consistent with standard Fe$_2$O$_3$

samples, which confirms the existence of the Fe species in a form of $Fe^{3+}$ species [16]. The catalyst has part of $ZnFe_2O_4$, so there is $Fe^{2+}$ inside (Figure 1), and in Figure 4a, the binding energy peak (B.E.P.) at 711.05, 714.82, 718.81 and 724.49 eV are associated with $Fe^{2+}$ and $Fe^{3+}$ species.

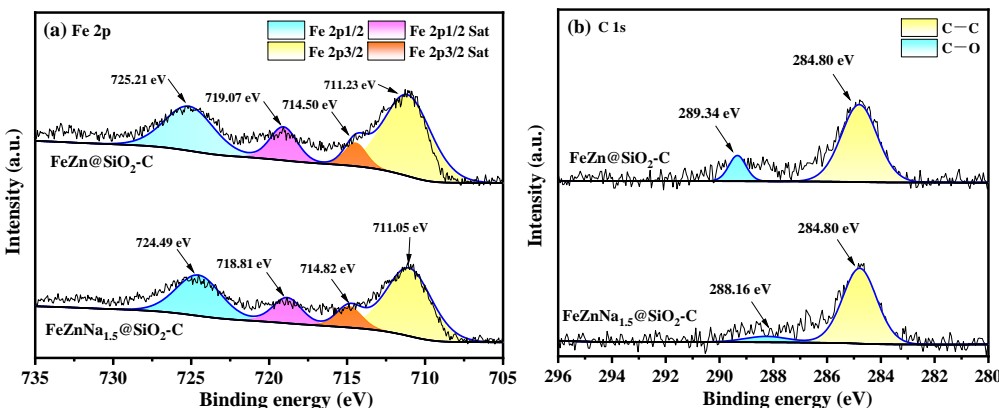

**Figure 4.** (**a**) Fe 2p and (**b**) C 1s XPS spectra of the $FeZnNa_x@SiO_2$-C catalyst.

### 2.1.3. Reduction Behavior

$H_2$-TPR experiments (Figure 5) showed that with Na doping in the $FeZnNa_{1.5}@SiO_2$-C, the peaks ascribed to $H_2$ consumption rose up to 550 °C, related to the reduction of the $Fe_2O_3$ to $Fe_3O_4$ and/or to FeO, and $Fe_2O_3$ to FeO and/or to $Fe^0$ shifted to a lower temperature, which suggests that the Na promoter concentration favored the initial reduction of the iron species [20]. On the contrary, the reduction peaks displaced towards high temperature for the $FeZnNa_{1.5}@SiO_2$-C, $FeZnNa_{3.1}@SiO_2$-C, $FeZnNa_{6.4}@SiO_2$-C and $FeZnNa_{7.6}@SiO_2$-C catalysts as the Na concentration increased. This displacement could be attributed to the closer contact between the iron oxides and potassium, as discussed in the previous XPS section. As for the peaks located at the high-temperature region (>550 °C), which were related to the reduction of $Fe^{3+}$ species to $Fe^0$. The noticed shift to high temperature and the increase in their intensities in $FeZnNa_x@SiO_2$-C catalysts could be attributed to a stronger interaction between the iron oxides and the $SiO_2$-C support, as suggested by the XPS results. Combined with TEM image (Figure 2), it can be found that the variation trend of $Fe_2O_3$ particle sizes in $FeZnNa_x@SiO_2$-C is almost the same as that of the $H_2$ reduction temperature.

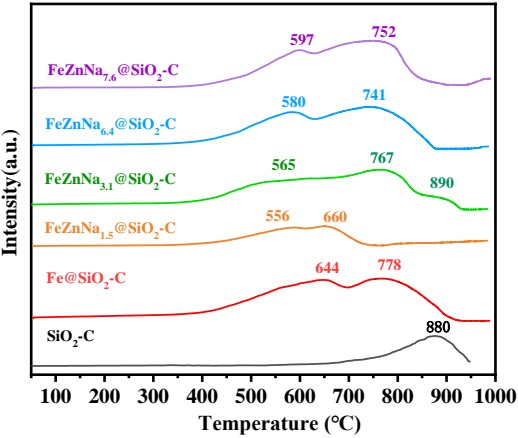

**Figure 5.** $H_2$-TPR profiles of the $FeZnNa_x@SiO_2$-C catalysts with different Na loading.

### 2.1.4. Chemisorption Behavior

The $CO_2$ adsorption properties of $FeZnNa_x@SiO_2$-C with different Na doping were further investigated by $CO_2$ temperature-programmed desorption ($CO_2$-TPD). As shown in Figure 6, three distinct peaks were observed for all the samples. The lower temperature

peak around 130 °C ($\alpha$ peak) was ascribed to the desorption of $CO_2$ weakly adsorbed in the bulk phase. The peaks in the temperature range of 300–400 °C ($\beta$ peak) and 500–700 °C ($\gamma$ peak) were due to the decomposition of $SiO_2$ or the desorption of $CO_2$, interacting strongly with the surface basic sites [36], which shift gradually to higher temperatures with increasing Na content. The increased Na loading also enhances the amount of $CO_2$ uptake at strong basic sites (Table 2) with the increased Na doping as well. The amounts of desorbed $CO_2$ for $\beta$ peak listed in Table 2 follow the sequence: $FeZnNa_{6.4}@SiO_2$-C > $FeZnNa_{3.1}@SiO_2$-C > $FeZnNa_{1.5}@SiO_2$-C > $FeZnNa_{7.6}@SiO_2$-C, which was consistent with the variation pattern of $CO_2$ conversion in Table 3. Considering that $FeZnNa_{1.5}@SiO_2$-C has a weaker $CO_2$ adsorption in the early stage, the $FeZnNa_{6.4}@SiO_2$-C catalyst has better $CO_2$ adsorption and support structure than other catalysts (Figure 2e), which may be related to the SSA enhancement and particle size reduction of the catalyst. The advantage of specific surface area also proved the adsorption of $CO_2$ was better, and the two characterizations could support each other.

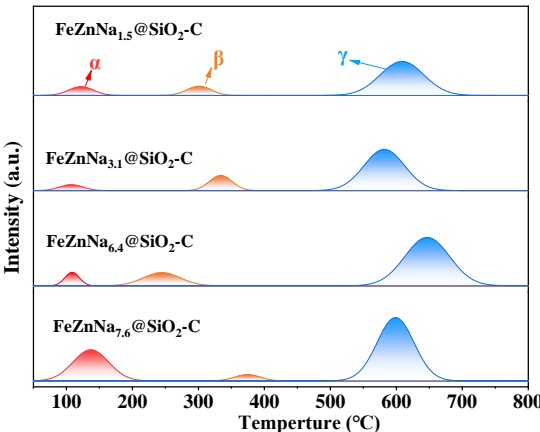

**Figure 6.** $CO_2$-TPD profiles of the $FeZnNa_x@SiO_2$-C catalysts with different Na doping in 50–800 °C.

**Table 2.** $CO_2$ uptake feature of the $FeZnNa_x@SiO_2$-C catalysts with different Na doping.

| Catalysts | The Amount of $CO_2$ Desorbed (µmol/g) | | | |
|---|---|---|---|---|
| | $\alpha$ Peak | $\beta$ Peak | $\gamma$ Peak | Total |
| $FeZnNa_{1.5}@SiO_2$-C | 58.5 | 60.4 | 396.4 | 515.3 |
| $FeZnNa_{3.1}@SiO_2$-C | 36.6 | 91.3 | 446.0 | 573.9 |
| $FeZnNa_{6.4}@SiO_2$-C | 51.4 | 128.6 | 576.7 | 756.6 |
| $FeZnNa_{7.6}@SiO_2$-C | 295.4 | 43.1 | 621.7 | 960.2 |

**Table 3.** Comparison of $CO_2$ hydrogenation Catalytic Performances: STY, selectivity of $CH_4$, $C_2$-$C_4$ olefins ($C_2^= - C_4^=$), $C_2$-$C_4$ paraffins ($C_2^0 - C_4^0$) and $C_{5+}$.

| Catalyst | $CO_2$ Con (%) | Selectivity [a] (%) | | | | O/P [b] | TOF [c] ($10^{-3}$ s$^{-1}$) | STY (g/g$_{cat}$·h) |
|---|---|---|---|---|---|---|---|---|
| | | $CH_4$ | $C_2^= - C_4^=$ | $C_2^= - C_4^=$ | $C_5^+$ | | | |
| $Fe@SiO_2$-C | 52.17 | 83.06 | 5.93 | 5.59 | 5.45 | 1.06 | 0.56 | 0.0057 |
| $FeZn@SiO_2$-C | 53.78 | 64.32 | 1.54 | 30.87 | 3.29 | 0.05 | 0.12 | 0.0015 |
| $FeNa@SiO_2$-C | 52.26 | 66.56 | 6.48 | 20.23 | 8.33 | 0.32 | 0.53 | 0.0090 |
| $FeZnNa_{1.5}@SiO_2$-C | 58.41 | 22.89 | 29.15 | 14.50 | 33.42 | 2.01 | 2.53 | 0.0315 |
| $FeZnNa_{3.1}@SiO_2$-C | 59.03 | 33.00 | 29.40 | 13.67 | 23.83 | 2.15 | 2.49 | 0.0321 |
| $FeZnNa_{6.4}@SiO_2$-C | 58.32 | 18.30 | 40.40 | 9.60 | 31.69 | 4.21 | 3.50 | 0.0436 |
| $FeZnNa_{7.6}@SiO_2$-C | 58.4 | 18.63 | 41.07 | 8.68 | 31.72 | 4.73 | 3.61 | 0.0400 |

Reaction condition: T = 300 °C, P = 2.5 MPa, $H_2/CO_2$ = 3, GHSV = 1500 mL/g·h, TOS = 100 h; Reduced in 5% $H_2$/Ar for 16 h at 400 °C before $CO_2$ hydrogenation. [a] Hydrocarbon selectivity was normalized with the exception of CO. [b] O/P is the molar ratio of olefin ($C_2^= - C_4^=$) to paraffin ($C_2^0 - C_4^0$). [c] TOF: turnover frequencies for $CO_2$ conversion, i.e., the number of $CO_2$ molecules converted per surface catalytic site per second at present $CO_2$ hydrogenation.

## 2.2. Catalytic Performances for CO$_2$ Hydrogenation

Table 3 shows the catalytic performance of this series of catalysts for CO$_2$ hydrogenation reaction. The ultimate goal of this series of catalysts is to improve the selectivity of olefin, so this section needs to focus on STY (space–time yield) of $C_2^= - C_4^=$. The catalysts discussed were divided into seven types, each of which was an average of multiple samples. Among them, the maximum value of STY was 0.0436 when the Na doping was 6.4%, and the observation of FeZnNa$_{7.6}$@SiO$_2$-C revealed that although the corresponding Na doping exceeded that of FeZnNa$_{6.4}$@SiO$_2$-C, the STY was relatively low, which shows that this series of catalysts are the best-quality catalyst when Na doping is less than 7.6%. The results of four catalysts with different Na doping (1.5–7.6%) were analyzed, and it was found that a further increase in Na doping did not improve the conversion rate of CO$_2$ any further, but rather suppressed the methane selectivity, improved the selectivity of C$_{2+}$, and significantly increased the value of low-carbon olefin STY. Compared with FeZnNa$_{1.5}$@SiO$_2$-C, FeZnNa$_{3.1}$@SiO$_2$-C and FeZnNa$_{7.6}$@SiO$_2$-C, FeZnNa$_{6.4}$@SiO$_2$-C exhibited higher CO$_2$ conversion, which probably had appropriate particle sizes and surface areas, providing more active sites and thereby increasing the CO$_2$ hydrogenation catalytic activity. Moreover, it is noteworthy in Table 3 that turnover frequencies of CO$_2$ conversion (TOF) for the FeZnNa$_x$@SiO$_2$-C catalyst ($3.61 \times 10^{-3}$ s$^{-1}$) is lower than that of FeSb ($3.6 \times 10^{-5}$ mol/g·s), and FeBi ($3.2 \times 10^{-5}$ mol/g·s) [37]. Although a high TOF value of $3.6 \times 10^{-5}$ mol/g·s was reported for FeSb, it should be noted that this value was obtained by the addition of the promoter Sb. All of these indicate that the catalysts (addition of promoters such as Na, Sb and Bi) can obviously promote TOF and STY, but Na was the lowest cost in all the above promoters.

## 2.3. Effect of Different Na Doping on Final Catalytic Performance

Figure 7 shows the effects of different Na-doping levels on the selectivity, CO$_2$ conversion rate and STY of CO, CH$_4$, $C_2^= - C_4^=$, $C_2^0 - C_4^0$ and C$_{5+}$ in the same series of catalysts. The CO$_2$ conversion has increased from about 54% to about 59% with the addition of Na, but the subsequent increase in Na doping has not increased significantly. With the increase in Na doping in the catalyst, the selectivity of the target product $C_2^= - C_4^=$ raised from 1.54% to 41.07%. STY achieved the maximum value when Na doping was 6.4%, which showed that although Na increased the activity of the catalyst, it inhibited the activity of the catalyst when the Na doping was too high. From above catalytic characterization, it can be found that FeZnNa$_{6.4}$@SiO$_2$-C had the most excellent performance with excellent mesoporous structure, highly dispersed metal nanoparticles, and the active component Fe$_5$C$_2$ (labeled as "●", Figure 8a) particles were also more obvious, and the metal and support had a strong interaction, which shows the advantages of accelerating the electron transport, improving the hydrothermal stability and catalytic activity, and the results of the characterization are consistent with the convincing experimental results. Furthermore, the synergistic effect of Na and Zn promoters not only increased the CO$_2$ adsorption amount, thereby promoting the complete transformation of Fe species into Fe$_5$C$_2$, but also reduced the Fe$_5$C$_2$ size and exposed more active sites, which was conducive to the FTO to consume CO and accelerated the conversion of CO$_2$. Therefore, the FeZnNa$_x$@SiO$_2$-C catalyst gave a higher CO$_2$ hydrogenation rate and $C_2^= - C_4^=$ formation rate.

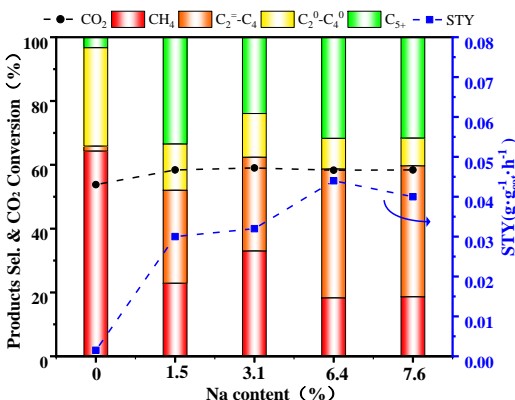

**Figure 7.** Effect of different Na doping on final catalytic performance.

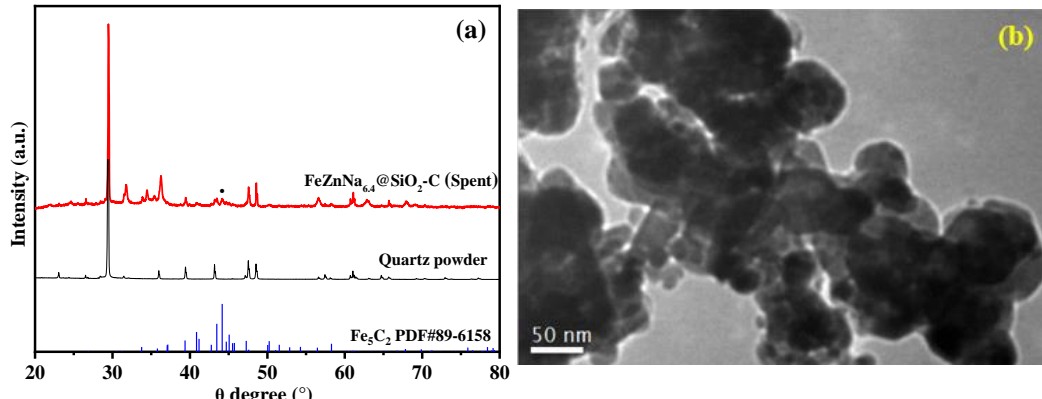

**Figure 8.** (**a**) XRD pattern and (**b**) TEM images of the spent sample-FeZnNa$_{6.4}$@SiO$_2$-C.

Figure 8b shows the TEM images of the spent sample; the metal particles had a slight enlargement, but did not congregate uncontrollably, which showed that graphitized carbon played a certain role, and when combined with the 100 h working condition diagram of Figure 9, it can be concluded that the catalyst has good structural stability.

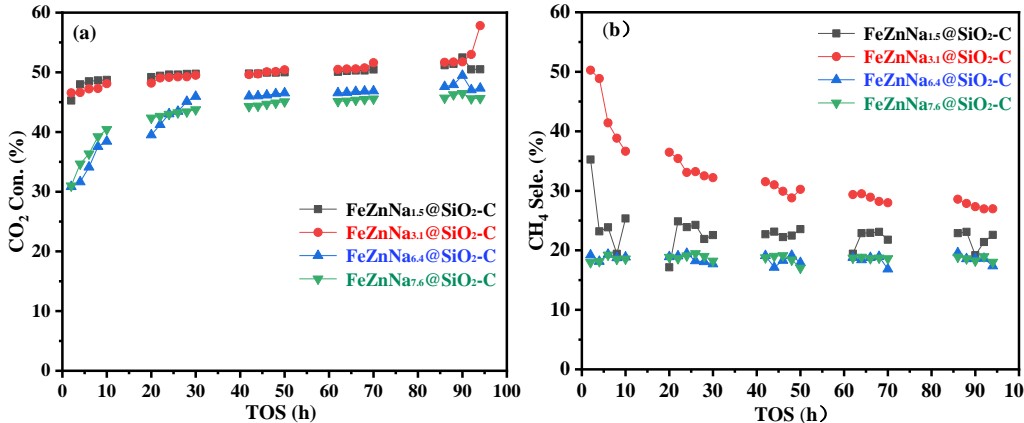

**Figure 9.** Change in (**a**) CO$_2$ conversion and (**b**) CH$_4$ selectivity with time on-stream for the FeZnNa$_x$@SiO$_2$-C catalysts.

XRD showed that iron-based catalysts containing Fe$_2$O$_3$ and ZnO components were successfully prepared. The approximate metal particle size (Fe$_2$O$_3$, ZnO) was calculated by the corresponding formula. The presence of Na and the proportion of components of each element were confirmed by XPS testing. From BET and TEM, it can be concluded that the FeZnNa$_x$@SiO$_2$-C catalysts were indeed mesoporous in structure. The pore size,

pore volume, specific surface area and precise metal particle size were also obtained; the high-resolution TEM diagram displayed the carbon layer wrapped on the surface of the catalyst and the highly dispersed metal nanoparticles. From the $H_2$-TPR test, the increase in Na doping significantly changed the reduction temperature of $Fe_2O_3$ species. The lowest reduction temperatures were observed for $FeZnNa_{6.4}@SiO_2$-C: 580 °C ($Fe_2O_3{\rightarrow}FeO$), 741 °C ($FeO{\rightarrow}\alpha Fe$). The initially prepared $Fe_2O_3$ phase was reduced to $Fe_3O_4$ at about 400 °C [11]. $CO_2$-TPD showed that more basic sites were provided for adsorption of more acidic $CO_2$ in the early stage due to the introduction of Na, which once again proved that the introduction of Na enhanced the $CO_2$ adsorption. In the later stage, support was decomposed with the increase in temperature. Compared with other catalysts, $FeZnNa_{6.4}@SiO_2$-C has a more stable support structure and $CO_2$ adsorption. Combining the above analyses, it is found that the reduction temperature, specific surface area, pore volume, average pore size and $CO_2$ adsorption are closely related to the particle size of $Fe_2O_3$. The reason for the change in particle size is due to the addition of alkaline elements (Na), Fe particle promotion, and the increase in metal particle size [34]. When Na reached a certain amount, Na acted as a dispersant and inhibited the aggregation of Fe particles, thereby reducing the particle size of $Fe_2O_3$. But as the Na doping increased again, the amount that acted as a dispersant was broken, and then the pore was blocked. Fe particles with a slightly larger particle size do not have extra positions to enter, and therefore the agglomeration phenomenon is produced again, and the particle size of Fe particles increases again. Therefore, the phenomenon of particle size change in $Fe_2O_3$ was first increasing, then reducing, and finally increasing. The reduction temperature, specific surface area, pore volume and average pore size are all affected by the $Fe_2O_3$ particle size and change accordingly. The specific internal connection is referred to above.

## 3. Experimental

### 3.1. Catalyst Preparation

$FeZnNa_{1.5}@SiO_2$-C was prepared by dissolving $Fe(NO_3)_3 \cdot 9H_2O$, $Zn(NO_3)_2 \cdot 6H_2O$, $NaNO_3$ and P123 in a mixed solvent formed by ethanol and deionized water and vigorously stirring for two hours to form a homogeneous solution. The mass of P123, $Fe(NO_3)_3 \cdot 9H_2O$, $Zn(NO_3)_2 \cdot 6H_2O$ and $NaNO_3$ were 2.5 g, 6.87 g, 4.36 g, 0.21 g, respectively. The volume ratio of ethanol deionized water in 40 mL mixed solvent was 1:1, and 4.46 mL of tetraethyl orthosilicate was added dropwise into the solution hydrolyzed in the previous step, and a homogeneous solution was formed by stirring for 1 h. After the end of the reaction, the solution was put into a reaction dish and placed in a fume hood for ventilation for 12–24 h, with the purpose of removing the surface ethanol and deionized water. The sample with the surface solvent removed were placed in an oven for heating and holding at 80 °C for 12 h. The sample was taken out and placed in a tubular furnace for calcination at 600 °C in air for 4 h at a rate of 2 °C/min to obtain the final catalyst material.

Two groups of catalysts were prepared according to the different Zn doping, namely $Fe@SiO_2$-C and $FeZn@SiO_2$-C; Fe accounted for 25% of the entire catalyst mass. Four groups of catalysts were prepared with Na doping as variables, with loading of 1.5%, 3.1%, 6.4% and 7.6%, respectively. Both Fe and Zn accounted each for 25% of the mass of the entire catalyst, named $FeZnNa_{1.5}@SiO_2$-C $FeZnNa_{3.1}@SiO_2$-C, $FeZnNa_{6.4}@SiO_2$-C and $FeZnNa_{7.6}@SiO_2$-C. Among them, the FeNa catalyst was additionally produced, named $FeNa@SiO_2$-C; Fe accounted for 25% of the mass of the entire catalyst, and Na accounted for 6.4% of the mass of the entire catalyst.

### 3.2. Nanocomposite Characterization

TEM images were obtained on a JEM-2010 electron microscope (JEOL (BEIJING) CO., Ltd., Beijing, China) with an acceleration voltage of 200 kV. XRD patterns were recorded on a Rigaku D/Max-2rB-II XRD meter (Cu Kα radiation, λ = 1.5406 Å, Rigaku, Tokyo, Japan), with 2θ = 10° to 90° scanned at 8°·min$^{-1}$. The texture properties of the nanocomposites were determined by $N_2$ adsorption/desorption isotherms on a Micromeritics ASAP 2010

apparatus (Micromeritics instrument (Shanghai) Ltd., Shanghai, China). The pore size distribution, pore volume and specific surface area were determined by the multipoint Brunauer–Emmett–Teller (BET) method. The concentration of surface metal atoms was determined using CO chemisorption in a Micromeritics ASAP 2010 (Micromeritics instrument (Shanghai) Ltd., Shanghai, China) automated system. A 0.1 g sample of a freshly calcined catalyst was first evacuated to $10^{-6}$ mm Hg at 100 °C for 30 min and then reduced under flowing $H_2$ by ramping at 2 °C/min to 280 °C and holding there for 12 h. The catalyst was then evacuated at 280 °C for 60 min to desorb $H_2$. CO chemisorption was carried out at 35 °C. X-ray photoelectron spectroscopy (XPS, Shimadzu (Shanghai) Experimental Equipment Co., Ltd., Shanghai, China) characterization was carried out in an UHV chamber equipped with a monochromatized Al K$\alpha$ X-ray source and a Sphere 2 analyzer. All the binding energies were calibrated by the adventitious C1s line at 284.8 eV. Hydrogen temperature-programmed reduction ($H_2$-TPR, BSD Instrument Technology (Beijing) Co., Ltd., Beijing, China) experiments were conducted on homemade apparatus equipped with a thermal conductivity detector ($CO_2$-TCD, BSD Instrument Technology (Beijing) Co., Ltd., Beijing, China). The reactor was loaded with 50 mg of sample at a time and heated to 1073 K at 10 K/min with a gas of 10% $H_2$/Ar.

### 3.3. Catalytic Reaction

It was used as a catalyst for the hydrogenation reaction of carbon dioxide in order to evaluate its catalytic performance. Firstly, the catalyst was placed in a mesh screen with the help of a grinding rod to obtain an experimental catalyst of about 0.5 g (40–50 mesh). Prior to $CO_2$ hydrogenation, 300 mg of catalyst was diluted with 600 mg of quartz powder (60–80 mesh) and reduced on site in flowing 5 vol % $H_2$/Ar (gas hourly space velocity (GHSV) = 2500 mL/g·h) at 400 °C for 18 h with a heating rate of 2 °C·min$^{-1}$. Catalytic testing was performed under industrially relevant operation conditions of 300 °C, 2.5 MPa, and a $H_2$/$CO_2$/$N_2$ ratio of 23/69/8 by volume (GHSV = 1500 mL/g·h) in a tubular fixed-bed reactor with an inner diameter of 10 mm. $N_2$ was used as the internal standard during gas-phase product analysis. All tubing and connections were wrapped with heating tape and heated at 150 °C to avoid condensation. The effluent of the reactor was analyzed on-stream by using a gas chromatograph (GC, Agilent 7890B, Agilent Technologies, Inc., Santa Clara, California, United States) equipped with a thermal conductivity detector (TCD, GC-2014AT, SHIMADZU, a TDX-01 column) and a flame ionization detector (FID, Agilent, a capillary column of PONA type). $H_2$, $N_2$, CO, $CH_4$ and $CO_2$ were detected by the TCD, hydrocarbons ($C_nH_m$), and the FID. To reduce the uncertainty in the product analysis, we collected the liquid products during every 12 h for one sample. The calculated carbon balance for all catalytic performance tests were between 95% and 105%. All listed data were obtained at 100 h on-stream when achieving the steady state.

The conversion of $CO_2$ ($CO_2$ conv.) and the selectivity of hydrocarbons ($C_nH_m$ sel.), CO (CO sel.) and $C_2^= - C_4^=$ ($C_2^= - C_4^=$ sel.) were calculated via the following equations:

$$CO_2 \text{ conv.}(\%) = \frac{CO_{2in} - CO_{2out}}{CO_{2in}} \times 100$$

$$CO \text{ sel.}(\%) = \frac{CO_{out}}{CO_{2in} - CO_{2out}} \times 100$$

$$C_nH_m \text{ sel.}(\%) = \frac{C_nH_m \times n}{\sum_1^n C_nH_m \times n} \times 100$$

$$C_2^= - C_4^= \text{ sel.}(\%) = \frac{\sum_2^4 C_nH_{2n} \times n}{\sum_1^n C_nH_m \times n} \times 100$$

$$\text{Carbon balance}(\%) = \frac{CO_{out} + \sum_1^n C_nH_m \times n + CO_{2out}}{CO_{2in}} \times 100$$

where $CO_{2in}$ and $CO_{2out}$ expressed the mole of $CO_2$, the inlet and the outlet of the reactor, respectively. $CO_{out}$ was the mole of CO in the outlet. $C_nH_m$ was the mole of the hydrocarbon product with carbon number n.

$$STY_{C_2^= - C_4^=} = \sum_2^4 C_nH_{2n} \times n \times M_{C_2^= - C_4^=}$$
$$TOF = \frac{CO_{2in} - CO_{2out}}{\omega_{Fe} \times D_{Fe} \div M_{Fe}}$$

where $STY_{C_2^= - C_4^=}$ is the time–space yield of $C_2^= - C_4^=$ based on unit mass catalyst $(g \cdot g_{cat}^{-1} \cdot h^{-1})$. TOF is the conversion frequency of catalyst $(h^{-1})$. $\omega_{Fe}$ is the mass fraction of Fe (%). $M_{Fe}$ is the relative atomic mass of Fe (56 g·mol$^{-1}$), and $D_{Fe}$ is the dispersion of Fe (%), which was based on total $CO_2$ chemisorbed, $CO_2/Fe_s = 0.5$, $n_{CO}/n_{Fe} = 0.5$, $D_{Fe}$ (%) = $2 \times n_{total\ CO\ chemisorbed} / n_{total\ number\ of\ Fe\ atoms}$.

## 4. Conclusions

The iron-based catalysts containing sodium and zinc additives have a stable mesoporous structure with graphitized carbon on the surface of the active component and are highly selective for the production of high-value olefins by $CO_2$ hydrogenation. In this paper, the $CO_2$ conversion of the FeZnNa catalyst was stable at about 59%, and the $CH_4$ selectivity was significantly inhibited compared with other catalysts, and its STY value was the highest 0.0436. The results showed that the presence of Na and Zn additives favored the carburizing of Fe species to generate smaller active $Fe_5C_2$ particles, improved the adsorption of carbon dioxide, and inhibited the deep hydrogenation of olefins, which is the key to the conversion of carbon dioxide into olefins. Catalysts with a sinter resistance and mesoporous resistance to collapse have more stable and long-lasting catalytic performance than other catalysts. The catalyst in this paper has sinter resistance and mesoporous collapse resistance, so it has a more stable structure than other catalysts. However, during the above reaction, the generated water inhibits the RWGS step, which reduces the conversion rate of $CO_2$ and hinders the formation of olefins in the FTO process, affecting the final catalytic performance. Therefore, the development of catalysts with hydrophobic surfaces to ensure the stability and durability of the catalytic performance is a difficult point for further development.

**Author Contributions:** Conceptualization, Z.N.; methodology, Z.N.; synthesis, Z.N. and M.C.; catalytic activity investigation, Z.N., M.C. and S.Z.; catalytic measurement preparation, M.C. and S.Z.; catalytic activity analysis and interpretation, Z.N. and X.C.; XRD investigation and data analysis, Z.N. and H.S.; $N_2$ physisorption measurements, X.C.; TEM investigation and interpretation, Z.N. and X.C.; TPR, TPD investigation and interpretation, Z.N. and L.S.; XPS data analysis and interpretation, Z.N. and M.C.; writing—original draft preparation, Z.N.; writing—review and editing, Z.N., M.C., S.Z., X.C., H.S. and L.S.; project administration, Z.N.; funding acquisition, Z.N. All authors have read and agreed to the published version of the manuscript.

**Funding:** The authors kindly acknowledge the financial support of National Natural Science Foundation of China (Grant No. 21905031) and Science Foundation of Changzhou university (ZMF18020299).

**Data Availability Statement:** The data presented in this study are openly available.

**Conflicts of Interest:** The authors declare no conflict of interest. The funders had no role in the design of the study; in the collection, analyses, or interpretation of the data; in the writing of the manuscript; or in the decision to publish the results.

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
