# Peer review of "Sodium Promoted FeZn@SiO2-C Catalysts for Sustainable Production of Low Olefins by CO2 Hydrogenation"

_catalysts, doi:10.3390/catal13121508_

Round 1
Reviewer 1 Report
Comments and Suggestions for Authors
This paper investigated the effect of Na and Zn in Fe-based catalysts for CO2 hydrogenation to low olefins. The authors used various characterizations to elaborate the different performance of those catalysts. However, the quality of manuscript needs to be improved. Some suggestions:
1. Please improve English used in this draft.
2. The author should provide the full name before using abbreviation, such as STY, RWGS, FTS, O/P.
3. Figure 1. Please add standard PDF of Na content you mentioned in the draft to show that there is no Na content peak in your catalysts.
4. Line 167. Is it Fe nanoparticles or iron oxide?
5. Figure 2. Those figures are unclear. Please provide good quality images.
6. Line 194-195. Why stronger CO2 adsorption would result in easier CO2 activation? Any reference?
7. Table 2. Only FeZnNa(6.4%) have Na/Fe ratio. How about others? Is it zero or no data?
8. Line 219-220. Can you point out which peak shifted to lower binding energy of catalysts with Na promoter? I didn't see any shift.
9. Line 224. How do you know O/Si ratio is higher that that of 2.0? Put data in the draft please.
10. Figure 4. Which specific catalysts did author do XPS? It shows FeZnNax, but there are 4 different Na percentage. Also, did author study how Na percentage affect bonding energy change by XPS?
11. Line 308. Authors mentioned enlargement of particles. Any size comparison to fresh one? Also, Figure 9 is not clear.
12. Figure 7. The left Y axis should be conversion instead of selectivity.
13. Line 309. For 100h stability test, authors only show stable selectivity. How about CO2 conversion?
14. Authors mentions Fe5C2 in abstract and conclusion. However, they never talked about it or show any evidence in main text.
Comments on the Quality of English LanguageThis manuscript is hard to read and understand. Please refine the whole draft regarding to following aspects:
1. Grammar and Syntax: There are frequent grammatical errors throughout the manuscript. Please review and revise for proper sentence structure.
2. Clarity and Coherence: Some sentences and paragraphs are unclear. Consider rephrasing to improve overall coherence.
3. Consistency: Maintain consistency in tense and point of view throughout the manuscript.
Some samples: Line 18-19, 35-37, 50-54, 88-92, 197-199.
Author Response
Dear reviewer,
Thank you very much for your letter and comments from reviewers. These comments are all valuable and very helpful for revising and improving our manuscript, and also provide important guidance to our research. We have studied the comments carefully and made corrections on the unsatisfying issues pointed out by the reviewers, which we hope will meet with your approval.
We have addressed the comments raised by the reviewers, and highlighted the amendments in red font in the revised manuscript. The point-by-point responses to the reviewers’ comments are listed below in this letter.
We appreciate for you and Reviewers’ warm work earnestly, and hope that the correction will meet with approval.
Thank you very much for your attention and consideration.
Sincerely yours,

Reviewer 2 Report
Comments and Suggestions for Authors
The manuscript by Ni et al. reports on a current and important topic, i.e. CO2 to olefins, however, there are some shortcomings in the manuscript that need to be addressed:
1. Details on the characterisation procedures of the catalysts need to be included in the Experimental Section.
2. Units for "airspeed" should be denoted as mL/g.h rather than mL/g*h
3. Tables should be denoted as Table 1, Table 2 etc in text rather than Tab. 1. "Tab" has a different meaning from "Table".
4. Table 1 - Values in this table should be written as whole numbers. The headings should indicate that these are crystallite sizes. The footnote should mention how the sizes were determined from XRD. Was it by the Scherrer equation?
5. Assuming the Scherrer equation was used to calculate crystallite size, how was this done for Fe2O3 in Fe@SiO2 and FeNa@SiO2? There are no peaks to be able to do this.
6. The resolution of the TEM images needs to be improved. The text on the images is not clear.
7. Table 2 - The surface areas are merely a repetition of data from Table 1. The PV and APS should be incorporated into Table 1.
8. XPS - Spaces are needed between the values reported in Line 229.
9. TPR - There is no discussion on the reduction of ZnO.
10. Table 3 - Is the C2-C4 column representing olefins or paraffins or both? This has not been communicated in the discussion. What is O/P and how is it calculated?
11. Table 4 - Again there is repetition from Table 3 (CH4 and C5+ selectivity). The data from both Tables should be combined with the C2-C4 olefins and paraffins reported separately.
12. Figure 7 - this is just Table 4 in graphical format. It can be excluded or can replace the tabulated data.
13. Figure 8 - What is the cause of the intense peak?
Comments on the Quality of English LanguageModerate editing of English language required
Author Response

(The authors gave the same response as above.)

Reviewer 3 Report
Comments and Suggestions for Authors
The authors did a series of characterizations to explain the catalyst activity, but the discussions were mostly hypotheses, which were not backed by strong compelling data. It was difficult to judge the paper due to the lack of control experiments. Below are my detailed comments.
For the equation in line 123, the authors should explain what each term is, or just write the Scherrer equation. Please explain which peak and diffraction angle was used to calculate the particle size. Also, show the peak deconvolution done to calculate the particle size.
Line 113: “the reaction conditions are 400 °C, 0.35 MPa,and the airspeed 113 is 2500 mL/(g*h).” What is airspeed? The reduction was not done under air!
Please provide the XRD for the following samples, SiO2-C, ZnO/SiO2-C, Na/SiO2-C, these will be good reference controls.
Please use ONLY TWO SIGNIFICANT figures for the data provided in the tables.
With the increase in Na loading the particle size does not increase significantly, so it will be good to not use strong adjectives like “significantly” in line 127.
Line 132-133: “With the increase of Na content, the particle size of Fe2O3 generally showed a trend of first rising, then decreasing, and finally rising”. Did the authors run a repeat to confirm this trend, if not then this claim is weak.
In Table 1 please provide metal weight loading for each catalyst, it will be convenient for the reader to refer to it while reading the article. Provide the SSA of SiO2-C.
It is important to provide the CO2 uptake (umol/ g) for these catalysts, obtained from CO2 TPD, to make claims like the one in Lines 185-186: “The smaller the size of the metal, the larger its surface area and the 185 more CO2 it adsorbs”. The reviewer expects to see these numbers.
The TEM image quality is low and the resolution is poor, please provide the particle size distribution chart, to make any claim about the average particle size. Can authors provide STEM-EDX for a few catalysts with high and low activity?
Why is the SSA data duplicated in both Tables 2 and 3?
Please provide DETAILED information on how each catalyst characterization was done.
Did the authors do air-free XPS for the catalyst? Authors do not have enough data to make this claim, “Among them, the existence of C-O bonds 222 indicates that there may be close interactions between C-Fe2O3-SiO2 in the form of C-O- 223 Fe/Fe-C-O or C-O-Si”, C-O can be from the adsorbed CO2 from the air, graphitic carbon, the XPS for reference samples are missing, without those data it difficult to agree with authors discussions and conclusions.
I have similar comments for H2-TPR, please provide the data for reference samples, like SiO2-C, FeOx/SiO2-C, etc. Without these data, it is difficult to appreciate the discussion. Also, for FeZnNa3.1 there is a small shoulder around 1200 K, what is this peak? Authors should also do a peak deconvolution to properly identify the reduction temp, rather than marking the maxima of the TPR profile.
Please provide the catalyst stability data, by providing the ToS data for CO2 conversion
What is the active site for this reaction, please provide the ToF for these catalysts and compare it with the other catalysts from literature.
Author Response

(The authors gave the same response as above.)

Round 2
Reviewer 1 Report
Comments and Suggestions for Authors
suggest to accept
Author Response
Thank you very much for your letter and comments from reviewers. These comments are all valuable and very helpful for revising and improving our manuscript, and also provide important guidance to our research. We have studied the comments carefully and made corrections on the unsatisfying issues pointed out by the reviewers, which we hope will meet with your approval.
The language of our manuscript has been refined again and polished by a professional editing company (LetPub). We have addressed the comments raised by the reviewers, and highlighted the amendments in blue font in the revised manuscript.

Reviewer 2 Report
Comments and Suggestions for Authors
In Table 1, SSA should be whole numbers and APV should have 1 decimal place.
Comments on the Quality of English LanguageMinor editing of English language required
Author Response

(The authors gave the same response as above.)

Reviewer 3 Report
Comments and Suggestions for Authors
No more comments
Author Response

(The authors gave the same response as above.)
